# Compact Image-Style Transfer: Channel Pruning on the Single Training of a Network

**DOI:** 10.3390/s22218427

**Published:** 2022-11-02

**Authors:** Minseong Kim, Hyun-Chul Choi

**Affiliations:** Intelligent Computer Vision Software Laboratory (ICVSLab), Department of Electronic Engineering, Yeungnam University, 280 Daehak-Ro, Gyeongsan 38541, Gyeongbuk, Korea

**Keywords:** image-style transfer, network pruning in a single training, channel loss, xor loss, computer vision, deep learning

## Abstract

Recent image-style transfer methods use the structure of a VGG feature network to encode and decode the feature map of the image. Since the network is designed for the general image-classification task, it has a number of channels and, accordingly, requires a huge amount of memory and high computational power, which is not mandatory for such a relatively simple task as image-style transfer. In this paper, we propose a new technique to size down the previously used style transfer network for eliminating the redundancy of the VGG feature network in memory consumption and computational cost. Our method automatically finds a number of consistently inactive convolution channels during the network training phase by using two new losses, i.e., *channel loss* and *xor loss*. The former maximizes the number of inactive channels and the latter fixes the positions of these inactive channels to be the same for the image. Our method improves the image generation speed to be up to 49% faster and reduces the number of parameters by 20% while maintaining style transferring performance. Additionally, our losses are also effective in pruning the VGG16 classifier network, i.e., parameter reduction by 26% and top-1 accuracy improvement by 0.16% on CIFAR-10.

## 1. Introduction

Recently, deep-learning-based image-style transfer methods [1,2,3,4,5,6,7,8] have achieved an impressive performance in generating an image of an arbitrary style. However, for the sake of this achievement, they used a common heavy network, a VGG feature network [9], which is designed to have a large number of parameters for general image-classification tasks with ImageNet [10]. This resulted in huge memory and computational power consumption in the feed-forwarding process of the network.

Another drawback of using a heavy network is overloading whitening and coloring transformer (WCT [3,4,11]), which transforms the second-order statistics of a feature map into that of target-style feature map. Singular value decomposition (SVD) of $O(n^{3})$ complexity is necessary for WCT, with a large number (n) of channels extracted from the VGG feature network being critical for style transferring speed. Therefore, using a compact network is necessary to improve the efficiency of image-style transfer in both memory usage and processing speed. Some studies [12,13,14,15,16] identified the channel redundancy of the VGG16 feature network [9] and this also shows the possibility of the channel pruning of the VGG19 feature network, which has more channels than the VGG16 feature network.

Network channel-pruning methods [12,13] remove channels of small filter weights in convolution layers. They follow a two-step process, first eliminating small magnitude filters from a trained network, and, second, re-training the reduced network to recover the possible degradation of performance due to the filter-removal step. These methods have the limitation of not considering the input magnitude coming into the convolution layer and this may result in occasionally omitting effective responses. For example, if an input feature map has large values, the convolution layer may generate responses of a sufficiently large magnitude even though the filter of a convolution layer has a small magnitude and these can affect the rear convolution layers.

In this paper, we propose a new channel-pruning method that automatically eliminates redundant convolutional channels during a single network training process without losing effective channels. For this purpose, we use two new losses, *channel loss* $L_{channel}$ and *xor loss*$L_{xor}$. $L_{channel}$ forces the output responses of redundant channels in the convolution layer to be zero. As the number of zero-response channels increases, the compactness of the network can be increased by eliminating the zero-response channels. $L_{xor}$ forces the zero-response channels to consistently appear regardless of the input image. This loss makes it possible to permanently remove zero-response channels without losing the performance of the network for an arbitrary input image. Once the consistent zero-response channels are obtained through an end-to-end network learning process with $L_{channel}$ and $L_{xor}$, filter parameters of convolution layers corresponding to the zero-response channels can be permanently removed, as shown in Figure 1. Since our pruning method is based on using additional losses which can be added to an original objective function, the network can be pruned in a single training process, unlike the previous channel-pruning methods which need multi-stages [12,13,14,15,16,17,18].

The main contributions of this paper are summarized as follows:Our *channel loss* increases the number of inactive channels, i.e., zero-response channels, of the feature map which increases the compactness of a network.Our *xor loss* forces a consistent position of zero-response channels regardless of the input image, which makes it possible to eliminate the corresponding filter parameters without losing the performance of the network.Our method achieved a compact network of 20% fewer parameters and 49% faster image-generating speed than the existing image-style transfer methods without performance degradation.Our method also achieved 26% fewer parameters and a top-1 accuracy improvement by 0.16% in the image classification task.

The rest of this paper is configured as follows. In Section 2, we will explain the existing methods of image-style transfer and channel pruning, and in Section 3, we will introduce the proposed pruning method. In Section 4, we will verify the effectiveness of the proposed method through appropriate experiments. Finally, we will conclude this work and discuss future work in Section 5.

## 2. Related Works

### 2.1. Image-Style Transfer

Gatys et al. [1] proposed a seminal work to transform the style of an image by using the VGG19 feature network [9]. They used the VGG19 feature network that was trained for image classification to extract the content and style features from an input image. The feature map from its deep layer was used as a content feature of the image and Gram matrices, which are correlation matrices of feature maps extracted from multiple layers, were used as a style feature of the image. Although this method can transfer the style of the content image to any target style, it takes a very long time to generate the stylized image due to the pixel-wise optimization process.

To deal with this slow-processing-time issue, some feed-forward network methods [19,20,21,22,23,24,25,26,27] were proposed to learn a VGG16-based or VGG19-based feed-forward network so that the stylized image should be quickly generated through a network forwarding pass. A feed-forward network learns to reduce the difference between the features [1] of the target images and the features of the generated image. Although these methods can quickly generate the stylized image, there exists a limitation that a style (or several styles) is fixed on the network.

To improve the style capacity of feed-forward networks, recent arbitrary style transferring methods [2,3,4,28,29] proposed quickly transfering an input image into an arbitrary style with a feature transformer. Huang and Belongie [2] extract the feature map from the image using the VGG19 feature network and then transform the mean and standard deviation of the feature map through adaptive instance normalization (AdaIN); then, the transformed feature map is decoded to a stylized image through a decoder network. The style prediction network [28] was also proposed for arbitrary style transfer. It predicts offsets and scale parameters. These parameters are used in the instance normalization layer. Although these methods can quickly generate the stylized image, there is a limitation that the correlation between channels of the feature map cannot be transformed.

To improve this limitation, recent methods [3,4] consider the channel correlation of feature maps using whitening and coloring transformer (WCT [3]), which has a burden of requiring a large amount of computational cost to calculate the square root and the inverse of the covariance matrix. Dynamic instance normalization (DIN) [29] was proposed for faster processing than WCT. DIN uses offsets and convolution weights predicted from two different networks, i.e., bias-net and weight-net, instead of the mean and covariance matrix of WCT. Since predicting these parameters is a simple feedforward process, using DIN achieved a very fast processing speed.

Liu et al. [30] proposed a new style-transform layer, AdaAttN, to use attention mechanisms for per-point style transfer. By utilizing the attention mechanism, they considered local similarities between input content and style images and applied them as the weights to calculate the mean and variance for AdaIN operation on each feature pixel.

### 2.2. Network Pruning

Recently, there have been proposed pruning methods [12,13] to accelerate network forwarding speeds and reduce the memory consumption of convolutional neural networks (CNNs) by eliminating the filter parameters of the convolution layer.

Li et al. [12] proposed a method to reduce the number of filters in convolution layers. This method measured the magnitudes of filters and removed the filter of the smallest magnitude. After this pruning process, this method performed long retraining (1/4 of initial training) to compensate for a possible performance reduction in the pruned network. Requiring an additional training procedure for the entire network is the disadvantage of this method.

He et al. [13] proposed a method to prune the filter channels of the convolution layer. To reduce some channels of the convolutional filter, this method learned a mask vector based on LASSO regression for selecting channels to be removed. After removing the selected channels by the learned mask vector, the remained convolution filters were retrained to reconstruct the original feature map. This method also requires two additional learning procedures, i.e., mask-vector learning for selecting channels to be removed and extra learning for original feature-map reconstruction.

A study for random channel pruning [31] has been proposed to benchmark the channel pruning methods so far. This study showed that repeating randomly selecting partial channels during the training phase achieved the best result. However, this method requires training many networks and also needs fine-tuning as an additional learning procedure after selecting top-N networks.

## 3. Method

In this section, we will explain how to reduce the number of channels in the feature map for channel pruning. In Section 3.1 and Section 3.2, we will introduce new losses for increasing the number of zero-response channels and for fixing the positions of zero-response channels in the feature map, respectively. The overall pruning process of applying our losses to the existing network learning procedure will be described in Section 3.3.

### 3.1. Channel Loss

The pruning methods based on the magnitude of filter parameters [12,13] have a limitation of not considering the magnitude of the input feature map. Therefore, we eliminate network parameters corresponding to zero-response channels regardless of the input magnitude through a network learning process to have a small number of nonzero-response channels in the feature map.

Let us consider a feature map $f^{l,b}\in R^{C^{l}\times H^{l}\times W^{l}}$ extracted from *l* th layer which has $C^{l}$ channels and spatial size ($H^{l},W^{l}$) of *b* th image in *B* batches. We define channel loss $L_{channel}$ as the number of non-zero-response channels in a feature map and this can be calculated as the sum of ${∥f_{i}^{l,b}∥}_{0}$ across channels and images:(1)$$L_{channel}=\frac{1}{B\cdot C_{l}}\sum_{b=1}^{B}\sum_{i=1}^{C_{l}}{∥f_{i}^{l,b}∥}_{0},$$
where ${∥x∥}_{0}$ is $l_{0}$-norm of *x* which is 0 when all elements of *x* are 0, or 1 otherwise. As $l_{0}$-norm is not differentiable and not suitable for back-propagation, we alternatively implement $l_{0}$-norm with the differentiable operations as Equation (Equation 2).
(2)$${∥x∥}_{0}=\frac{{∥x∥}_{2}}{{∥x∥}_{2}+\epsilon },$$
where ${∥\cdot ∥}_{2}$ represents $l_{2}$-norm and the small value $\epsilon =1\times 10^{-5}$ is used to avoid divide-by-zero.

### 3.2. XOR Loss

Although the feature map has a large number of zero-response channels by teaching the network to reduce $L_{channel}$, we cannot permanently eliminate the zero-response channels because their positions vary from image to image. To fix the positions of the zero-response channels, we introduce XOR loss $L_{xor}$ (Equation (Equation 3)), to quantitatively calculate the positional variation of the zero-response channels, as in Figure 2.
(3)$$L_{xor}=\frac{2}{B\left(B-1\right)\cdot C_{l}}\sum_{i=1}^{B}\sum_{j=i+1}^{B}\sum_{t}^{C^{l}}\left|{∥f_{t}^{l,i}∥}_{0}-{∥f_{t}^{l,j}∥}_{0}\right|,$$
where *B* is the number of batch images and $f_{t}^{l,i}$ represents the *t* th channel in the feature map of the *l* th layer from the *i* th image. We need to use a sufficiently large number of images for representative position consistency by $L_{xor}$. For the style transfer task, we use all input images (B=8) during the network training phase [3,4] and, thus, 28 (=${}_{8}C_{2}$) pairs of feature maps are used in calculating $L_{xor}$. In addition, for the image classification task, we use all input images (B=128) during the network training phase [32] and, thus, 8128 (=${}_{128}C_{2}$) pairs of feature maps are used during the network training phase. We found each optimal batch size through several experiments.

### 3.3. Channel Pruning during Target Task Learning

Our losses are for reducing the number of channels in the feature map regardless of the target task of a network. Therefore, they can be applied to the network learning process very simply using the total loss $L_{total}$ (Equation (Equation 4)), the weighted sum of our losses and the original target loss $L_{target}$.
(4)$$L_{total}=L_{channel}\cdot \lambda_{channel}+L_{xor}\cdot \lambda_{xor}+L_{target}$$

For example, applying our pruning losses to the recent style transfer networks [3,4] can be carried out by adding $L_{channel}$ and $L_{xor}$ calculated in a batch to their original reconstruction losses.

We used $\left(\lambda_{channel},\lambda_{xor}\right)=\left(1.0,1.0\right)$ for the style transfer task and (0.005,0.005) for the image classification task, which are found through several experiments to prune the channel while maintaining the sufficient performance of the task.

After teaching a network to reduce $L_{total}$, the network is optimized to perform the target task and to have many consistent zero-response channels in the feature map simultaneously. Since these zero-response channels do not mathematically affect the operation of the next convolution layer, after training a network, the convolution filters that generate these zero-response channels and the convolution filters that take these zero-response channels as input can be eliminated from the network, as shown in Figure 1.

## 4. Experiments

### 4.1. Experimental Setup

In this section, we describe each experiment setting (image style transfer, image classification) that we performed to verify the effectiveness of the proposed method. As a common setting, all experiments were performed in the environment of NVIDIA GTX 1080 TI GPU and Pytorch framework with CUDA and cuDNN libraries.

#### 4.1.1. Setup for Image Style Transfer

We used MS-COCO train2014 [33] as a content image dataset and the training dataset of Painter By Numbers [34] as a style image dataset to train the image-style transfer networks [3,4]. Each image of the dataset was resized into 512 pixels on the shorter side while maintaining the original aspect ratio. During the network learning process, the image was randomly cropped into 256×256 pixels to avoid boundary artifacts. We used an input batch size of 8 and Adam optimizer [35] with a learning rate of 1 × 10^−4^ during a total of 40,000 iterations for network training. We used MS-COCO test2014 [33] and test dataset of Painter By Numbers [34] for quantitative experiments. We performed all experiments with the same network structure as the existing style transfer networks [3,4].

#### 4.1.2. Setup for Image Classification

We used the VGG16 classifier network [9] for the efficiency of the experiment, where it has fewer parameters than the recently proposed networks [36,37,38], and used CIFAR-10 dataset [39] consisting of a small number of images. Since the VGG16 classifier network was designed to classify ImageNet [10] with 1000 classes, we modified the network for CIFAR-10 dataset with 10 classes [32]. We used the SGD optimizer [40] with a batch size of 128 for 300 epochs to train a network and used the CIFAR-10 test dataset to measure the performance of the trained network. The other options not mentioned here were set to the same as in [32].

### 4.2. Experimental Results of Pruning for Image-Style Transfer Task

In this section, we compare the performance of the recent two style transfer networks (Universal [3], AvatarNet [4]) and the result of our pruning method. Note that while both methods [3,4] learn only the decoder, we learn both encoder and decoder to remove the channel of the encoded feature map and to learn the original target task using an end-to-end learning scheme. Therefore, to compare the performance of the proposed method fairly, we also compared the end-to-end learning results [41] of two style transfer networks without pruning.

#### 4.2.1. Analysis of Feature-Map Channel Response

To verify if the style transfer network trained by our method extracts a feature map with consistent zero-response channels for arbitrary input images, we analyzed the channel responses of feature maps extracted from 1000 unseen test images. Figure 3 shows the non-zero/zero-responses of feature maps extracted from all 1000 test images with several single-scale transfer networks. As shown in Figure 3a, the feature map extracted by the VGG16 [9] feature extractor used in the existing style transfer methods [2,3,4] shows full non-zero-responses (white region). In Figure 3b, feature maps extracted by the uncorrelated encoder [42] show a number of zero-response channels, but their indices vary from image to image. The zero-response-channel indices of our method with both channel loss (Equation (Equation 1)) and xor loss (Equation (Equation 3)) are consistent through almost all images. Due to the consistent indices of zero-response channels, we can eliminate the convolution filter parameters corresponding to the zero-response channels, where they do not affect the output of the next convolution layer.

#### 4.2.2. Efficiency in Memory and Speed

Table 1 shows the number of parameters for each network and the average (standard deviation) processing time (ms) for each element calculated using 1000 test images. Results of the end-to-end learning scheme [41] are not compared here because they use the same number of parameters as existing methods [3,4].

As we can see from the speed measurements in Table 1, most of the image generation time is consumed by the transformer. As we mentioned in Section 1, both methods use WCT [3], which requires $O(n^{3})$ complexity to the number (n) of feature map channels. Therefore, we removed the redundant channels in the convolutional layer to improve the style transferring speed of the existing methods, and, as a result, we were able to shorten the transformer time of Universal by 49% (Table 1 (b)) and AvatarNet by 39% (Table 1 (d)), respectively. We reduced the number of parameters for Universal and AvatarNet by 20% and 24%, respectively, by removing the filters that generate the zero-response channels and the filters that receive the zero-response channels as input.

#### 4.2.3. Quality of Stylized Image

Here, as we mentioned in Section 4.2, for a fair comparison, we further compared end-to-end learning results [41] of the existing methods [3,4] and the result of the proposed method.

By learning the style transfer networks with an end-to-end learning scheme [41], we can see that the color tone matching of the output image is improved (Figure 4 from (a) to (b) and from (d) to (e)), as indicated by Yoon et al. [41]. For example, if we compare the fifth row of Figure 4a,b, we can see that some color of the flower is not completely transformed into the color tone of the style image in the output image of the existing method (Figure 4a), but it is completely transferred in the output image of end-to-end learning result (Figure 4b). In addition, comparing the second row (Figure 4d,e), we can see that the various colors (white and red) of the style image are completely transferred in the output image of the end-to-end learning scheme (Figure 4e) compared with the existing method (Figure 4d). In addition, the output image (Figure 4c,f) of the proposed pruning method shows that there is no serious quality deterioration in the output image even though it uses 20% and 23% fewer parameters than the existing methods (Figure 4b,e).

#### 4.2.4. Comparison with the Existing Pruning Method

In this section, we compare our pruning method with the existing pruning method. Among the two previous pruning methods [12,13], we selected Li et al. [12] as the comparison method since it prunes network during its encoder training phase in the same way as our method, while He et al. [13] requires an additional training procedure for mask learning and decoder learning to reduce the feature-map channels. We applied the existing filter pruning method [12] to the recent image style transfer networks [3,4] and compared the result with the result of our pruning method. For the pruning process of the existing method [12], a learned-style transfer network [3,4] was pruned to have the same number of channels as our pruning method and then retrained to avoid possible performance degradation by 10,000 iterations (1/4 of initial learning iterations [12]). We also obtained the result of end-to-end learning [41] as a baseline of the encoder learning scheme without pruning.

Comparing the results of the proposed method (Figure 5b,e) with the end-to-end learning results (Figure 5a,d) in the enlarged area of Figure 5, we can see that the output image of the proposed method has no serious degradation.

In contrast, as we can see in the lips of the magnified image, the output images of the existing pruning method were not completely generated in the aspect of color tone (Figure 5c) and detailed edges (Figure 5f) of the style image. In addition, the result of our pruning method achieved 17% lower style loss [1] for 1000 test images than the existing pruning method.

Figure 6 shows the style and content losses of the generated images correspond to the number of removed channels in a network. The horizontal axis of the graphs represents the number of removed channels and the vertical axis represents the content or style loss in log scale. For the previous filter pruning method [12], we retrained the networks after removing 25×N channels (N=0…10) and measured each loss of images from the networks. For our method (red line in Figure 6), since it does not require an additional learning process after channel elimination, we measured each loss as gradually removing the redundant channels from an initially trained network. We selected the channels of the smallest cumulative magnitude for the entire training dataset [33,34] as the redundant channels.

Since the network pruned by our method has many non-zero-response channels (Figure 3c), the content and style losses do not increase until the removal of about 200 channels, as shown as red lines in Figure 6a,b. When the number of the removed channels exceeds 200, the losses increase sharply because the non-zero-response channels of the feature map are removed. In contrast, the previous filter pruning method [12] shows gradually increasing losses as the number of removed channels increases (blue lines in Figure 6a,b. When the number of the removed channels exceeds 200, it performs better (lower losses) than our method because of its network retraining process. For a fair comparison, we performed retraining after channel removal for our method, as with the previous method [12]. After retraining, our method has fewer losses than the previous method, as shown as the black lines in Figure 6a,b. Therefore, we can say that our pruning method achieved better loss performance than the previous method of up to 200 channel removal without retraining, and still better with retraining after removing more than 200 channels.

### 4.3. Experimental Results of Pruning for Image Classification Task

Table 2 shows the performance of the VGG16 classifier network (a), the result of the proposed pruning method (b), and the result of the existing pruning method [12] (c) on CIFAR-10. We learned the network with proposed pruning losses using the feature maps of reluX_1(X=1,…,5) and were able to reduce the number of parameters as shown in Table 2 (b). In addition, the existing pruning method (Table 2 (c)) which can select the number of parameters to be removed pruned trained network (a) to have the same number of parameters as ours (b) and then re-trained the pruned network for the possible degradation of classification performance during 75 epochs (1/4 of initial training epochs).

Since the proposed method removes the redundant channel of the network to have the optimal number of parameters through learning, the overfitting to the training data is reduced and, thus, the performance of the test dataset is improved by 0.16% over the existing network (Table 2 (a)) composed of redundant parameters [12,13,14,15,16]. Moreover, since the proposed method selects and removes the redundant channels optimally through a network training process, it achieved an accuracy improved by 0.46% (Table 2 (c)) over the existing method [12] which removes channels of small magnitude where these channels might be effective in style transfer.

## 5. Conclusions

In this paper, we proposed a one-step pruning method which removes the redundant channels of the convolutional layer by adding the proposed two pruning losses to an original objective function of the network. The proposed $L_{channel}$ forced the output responses of the redundant channels in the convolutional layer to be zero. In addition, $L_{xor}$ forced these zero-response channels to consistently appear in the same position regardless of the input image. Based on our experiments, we could not obtain any pruning effect using one of these two losses but only using both losses. Therefore, by teaching the network to reduce the proposed two losses, we were able to increase the compactness of the network by eliminating the zero-response channels. By applying the proposed method to the recent style transfer networks, we were able to reduce the parameters of existing networks by up to 20% and generate images 49% faster without hurting performance deterioration. In addition, we applied the proposed method to the image classifier network and reduced the parameters of the network by 26% with a top accuracy improvement of 0.16%. Our pruning losses have the advantage of reducing the channel by simply adding it to the original objective function, but there is a limitation in that the number of channels to be removed cannot be manually selected. Therefore, we are planning an attempt to improve the controllability of the proposed pruning method.

## Figures and Tables

**Figure 1 sensors-22-08427-f001:**
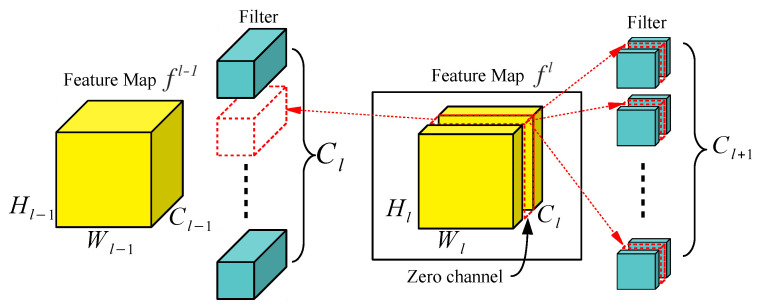
Removing filters based on the zero-response channel in the feature map: by removing the zero-response channel of the feature map $f^{l}$, it is possible to reduce the filters of the encoder convolution layer that generates the zero-response and the filters of the decoder convolution layer that takes the zero-response as input.

**Figure 2 sensors-22-08427-f002:**
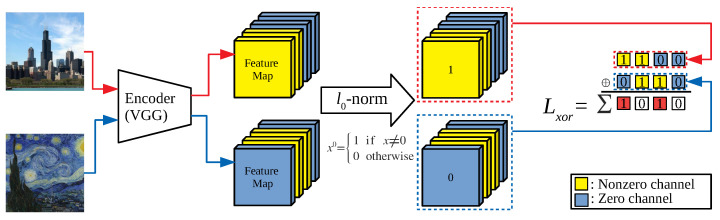
The process of calculating xor loss: The quantitative positional variation in zero-response channels can be measured by counting the number of consistent zero/non-zero-response channel through images.

**Figure 3 sensors-22-08427-f003:**
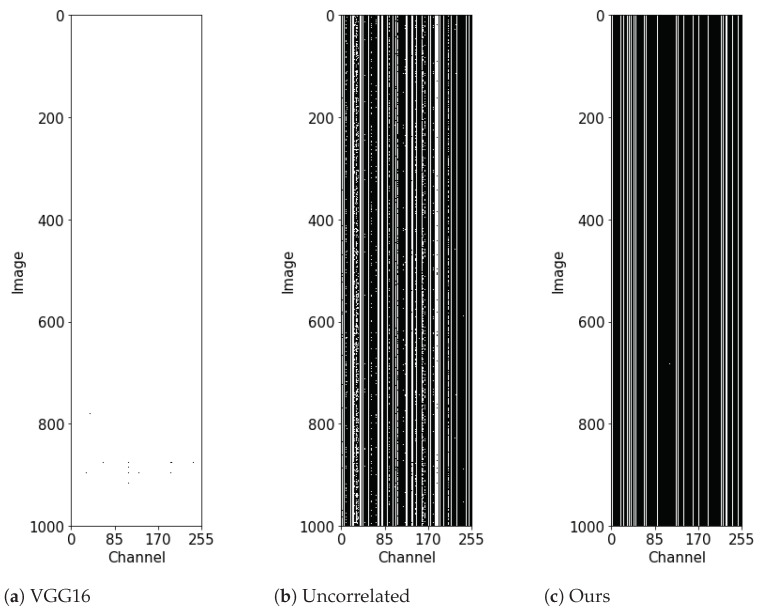
Channel responses of feature maps extracted from 1000 test images: The horizontal and vertical axes represent channel indices and image indices, respectively. The feature map extracted from our pruned network (**c**) shows only 70 non-zero-response channels (white regions) out of a total of 256 channels. Compared to the original VGG16 [9] (**a**) or uncorrelated encoder [42] (**b**), our channel responses show a larger number of consistent zero-response channels.

**Figure 4 sensors-22-08427-f004:**
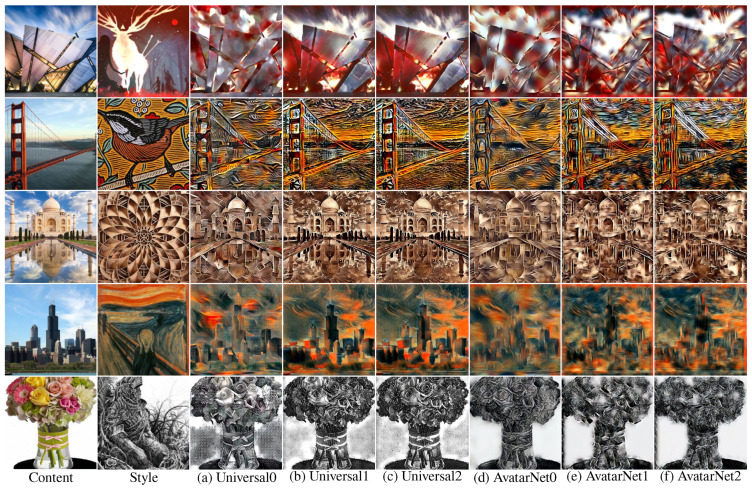
Comparison of the output image of existing style transfer methods (Universal [3] (**a**), AvatarNet [4]) (**d**), results of the end-to-end learning scheme [41] (**b**,**e**), and the results of our method (**c**,**f**): the output images were generated using the images not used during the network learning.

**Figure 5 sensors-22-08427-f005:**
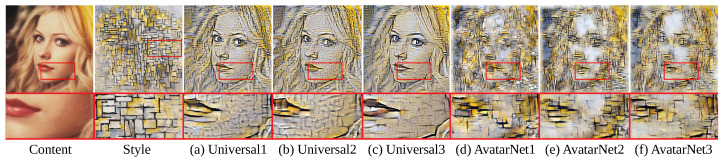
Comparison of the output images of pruned style transfer network using the proposed pruning method and the existing pruning method [12]: the existing method used the same number of parameters as the proposed method. (**a**) Universal [3] with end-to-end learning scheme [41], (**b**) Universal with our pruning, (**c**) Universal with the previous pruning [12], (**d**) AvatarNet [4] with end-to-end learning scheme [41], (**e**) AvatarNet with our pruning, (**f**) AvatarNet with the previous pruning [12].

**Figure 6 sensors-22-08427-f006:**
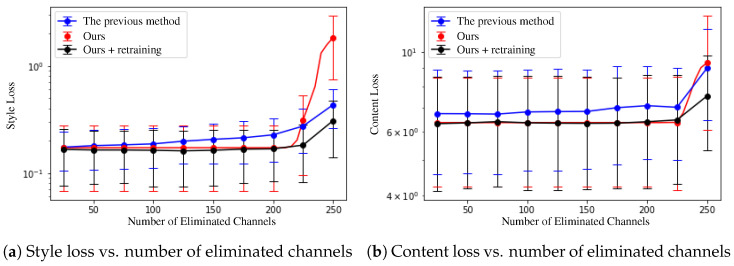
Loss variation corresponding to the number of eliminated channels. The blue lines are the results of the previous pruning method [12].

**Table 1 sensors-22-08427-t001:** Quantitative comparison of existing style transfer methods and the result of the proposed pruning method: the speed (ms) is the average (standard deviation) measured using 1000 test images not used during the network learning.

Methods	Speed (ms)			Memory
	Encoder/Decoder	Transformer	Total	(# of Parameters)
(a) Universal	6.67 (0.05)	377.80 (5.26)	384.47 (5.29)	34 M
(b) Universal + ours	6.76 (0.07)	190.20 (3.83)	196.95 (3.93)	27 M
(c) AvatarNet	2.93 (0.07)	325.53 (7.02)	328.46 (7.05)	7 M
(d) AvatarNet + ours	2.84 (0.08)	198.97 (12.43)	201.80 (12.52)	5 M

**Table 2 sensors-22-08427-t002:** Classifier pruning results using the proposed pruning method and existing pruning method [12]: the existing pruning method (c) was pruned to have the same number of channels as the result of the proposed method (b).

	# of Parameters	Top-1 Error (%)
(a) Base	15 M	7.74%
(b) Ours	11 M	7.58%
(c) Li et al. [12]	11 M	8.04%

## Data Availability

Not applicable.

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
