# Peer review of "Compact Image-Style Transfer: Channel Pruning on the Single Training of a Network"

_sensors, 2022, doi:10.3390/s22218427_

Round 1

Reviewer 1 Report

The manuscript is well organized. I enjoyed reading it. However,  the figures' resolution must be more clear and more explained.

Author Response

As the reviewer’s concern, we found that figure 3 and 6 are in low resolution and need to be improved in resolution. Therefore, we modified figure 3 and 6 to have higher resolution for more clear understanding in our revised manuscript.

Reviewer 2 Report

This work proposes an algorithm for Compact Image Style Transfer. The application is important, but there are several major concerns need to be addressed:

1- This work is mainly based on previous works. The technical novelty of this work is very limited. Ideally a lot more technical novelty is expected.

2- The authors should better explain how the model parameters are tuned.

3- There should be a more detailed ablation study presented in this paper. 

4- The style transfer technique is very important in augmented reality effects, and the authors should emphasize this.

5- Many of the recent relevant works, and also AR related works are missing in introduction, discussions, and references. The authors should do a more detailed literature overview and add more works. Some of them are suggested below:

- "Adaattn: Revisit attention mechanism in arbitrary neural style transfer." Proceedings of the IEEE/CVF international conference on computer vision. 2021

- "Modern Augmented Reality: Applications, Trends, and Future Directions." arXiv preprint arXiv:2202.09450 (2022).

6- There are several grammatical errors in this manuscript. Please do a proofread and fix them all.

Author Response

1. We had already shown the contribution of our work on page 2 in the introduction section of the original manuscript as follows:

The main contributions of this paper are summarized as follows:

1. Our channel loss increases the number of inactive channels, i.e., zero-response channels, of the feature map which increases the compactness of a network.

2. Our xor loss forces a consistent position of zero-response channels regardless of the input image which makes it possible to eliminate the corresponding filter parameters without losing the performance of the network.

3. Our method achieved a compact network of 20 % fewer parameters and 49 % faster image-generating speed than the existing image style transfer methods without performance degradation.

4. Our method also achieved 26 % fewer parameters and a top-1 accuracy improvement of 0.16 % in the image classification task

One of our main technical novelty is proposing both channel loss and xor loss for channel pruning. By using these losses together, we enforce the encoder to have consistent zero-response channels as many as possible in the training phase. Then we can prune the network by eliminating the zero-repose channels. This pruning during a single training phase is totally new method without any previous base.

Another novelty of our work is high pruning efficiency in two representative computer vision applications, i.e., image style transfer (20 % fewer parameters) and image classification task (26 % fewer parameters).

2. We had already mentioned how to train the network parameters in the method and experimental setup sections of our original manuscript as follows:

Line 163 – 165: “We used (λchannel , λxor) = (1.0, 1.0) for the style transfer task and (0.005, 0.005) for the image classification task, which are found through several experiments to prune the channel while maintaining the sufficient performance of the task.”

Line 178 – 187: “We used MS-COCO train2014 [33] as a content image dataset and the training dataset of Painter By Numbers [34] as a style image dataset to learn the image style transfer networks [3 ,4]. Each image of the dataset was resized into 512 pixels on the shorter side while maintaining the original aspect ratio. During the network learning process, the image was randomly cropped into 256 × 256 pixels to avoid boundary artifacts. We used an input batch size of 8 and Adam optimizer [35] with a learning rate of 1e-4 during a total of 40,000 iterations for network training. We used MS-COCO test2014 [33] and test dataset of Painter By Numbers [34] for quantitative experiments. We performed all experiments with the same network structure as the existing style transfer networks [3,4].”

Line 188 – 196: “We used the VGG16 classifier network [9] for the efficiency of the experiment, where it has fewer parameters than the recently proposed networks [36–38], and used CIFAR-10 dataset [39] consisting of a small number of images. Since the VGG16 classifier network was designed to classify ImageNet [10] with 1,000 classes, we modified the network for CIFAR-10 dataset with 10 classes [32]. We used the SGD optimizer [40] with a batch size of 128 for 300 epochs to train a network and used the CIFAR-10 test dataset to measure the performance of the trained network. The other options not mentioned here were set as the same in the [32].

3. We had already tested with only xor loss or only channel loss. As a result, using only one of the losses was not effective in pruning. Only with both xor and channel losses, we could get figure 3(c) in that both consistent and sparse non-zero channels. Therefore, we did not add an ablation study for each loss but an ablation study with no additional loss (figure 3(a)), uncorrelation loss (figure 3(b)), and our losses (figure 3(c)).

In figure 6, in addition, we had already presented an ablation study about retraining in our original manuscript. We compared a two-stage learning scheme [7], our single learning scheme (ours), and ours + retraining.

4. Our work is focusing on a new pruning technique during the network learning procedure. And we selected two applications, i.e., image style transfer and image classifier, to show the effectiveness of our pruning method. So, even if image style transfer is important to augmented reality, emphasizing augmented reality may distract the main topic of our manuscript, network pruning.

5. We carefully inspected the suggested references and found that ‘AdaAttN’ is closely related to our work. I am afraid that augmented reality is not related to the main topic of our work. We reviewed ‘AdaAttN’ as a recent related work and put some description about this into the related works section of our revised manuscript.

Liu et al. [30] proposed a new style transform layer, AdaAttN, to use attention mechanism for per-point style transfer. By utilizing the attention mechanism, they considered local similarities between input content and style images and applied them as the weights to calculate mean and variance for AdaIN operation on each feature pixel.”

6. As the reviewer’s comment, we found several grammatical errors in our original manuscript. We fixed them and proofread the manuscript once again.

Reviewer 3 Report

Authors have presented their work with Paper Title: "Compact Image Style Transfer: Channel Pruning on a Single Training of Network "

The Authors contribution of this paper are summarized as follows:

Channel loss increases the number of inactive channels, i.e., zero-response channels, of the feature map which increases the compactness of a network.

XOR loss forces a consistent position of zero-response channels regardless of input image which makes it possible to eliminate the corresponding filter parameters without losing the performance of the network.

The proposed method achieved a compact network of 20 % fewer parameters and 49 % faster image generating speed than the existing image style transfer methods without performance degradation.

Their method also achieved 26 % fewer parameters and top accuracy improvement of 0.16 % in the image classification task.

I request the authors to present few more latest methods comparison with your proposed method in terms of Image data analytics.

Author Response

As the reviewer’s comments, we reviewed a more recent work related to image style transfer as below.

-[30] "Adaattn: Revisit attention mechanism in arbitrary neural style transfer." Proceedings of the IEEE/CVF international conference on computer vision. 2021

And we put some description about this into the related works section of our revised manuscript.

Liu et al. [30] proposed a new style transform layer, AdaAttN, to use attention mechanism for per-point style transfer. By utilizing the attention mechanism, they considered local similarities between input content and style images and applied them as the weights to calculate mean and variance for AdaIN operation on each feature pixel.”